biochemistry/chemical biology/materials science

modified polysulfone, sulfonated citric chitosan, graft modification, hydrophilicity, haemocompatibility

**Author for correspondence:**
Yunren Qiu
e-mail: csu_tian@csu.edu.cn

This article has been edited by the Royal Society of Chemistry, including the commissioning, peer review process and editorial aspects up to the point of acceptance.

# Preparation of modified polysulfone material decorated by sulfonated citric chitosan for haemodialysis and its haemocompatibility

Bingxian Lin, Kaiming Liu and Yunren Qiu

College of Chemistry and Chemical Engineering, Central South University, Changsha 410083, People's Republic of China

BL, 0000-0001-7064-9081; YQ, 0000-0003-1705-5051

Polysulfone (PSF) works potentially in haemodialysis due to its great mechanical and chemical stability, but performs poorly in haemocompatibility. For promoting the unpleasant haemocompatibility, sulfonated citric chitosan (SCACS) with the structure and groups similar to heparin was primarily synthesized by acylation and sulfonation. Furthermore, the chloroacylated PSF was pretreated by electrophilic chloroacetyl chloride to achieve more active sites for further reaction; the following membranes underwent the amination and were named amination polysulfone (AMPSF) membranes. Moreover, SCACS with abundant carboxyl and sulfonic groups was covalently grafted at the surface of pretreated PSF membranes, called PSF-SCACS membranes. The PSF-SCACS membranes were successfully synthesized and characterized by $^1$H NMR, ATR-FTIR and XPS. In addition, the water contact angle of PSF-SCACS membranes decreased by 47° and the morphologies of the membranes changed little compared with the unmodified PSF membranes. The haemocompatible testing results, including protein adsorption, platelet adhesion, haemolysis rate, plasma recalcification time, activated partial thromboplastin time (APTT), prothrombin time (PT) and thrombin time (TT), demonstrated that the PSF-SCACS membranes possessed excellent haemocompatible performances, and SCACS played an important role in the modification.

## 1. Introduction

At present, the population undergoing haemodialysis treatment has increased at a rate of more than 11% per year, while the incidence of complications caused by acute and chronic renal

failure is accelerating year by year [1]. As a result, the haemodialysis treatment is more and more widely applied in clinical. Haemodialysis membranes as the most main part of haemodialysis are studied extensively, which eliminate toxins from the blood and maintain the electrolyte balance in the blood [2].

The development of haemodialysis membranes has gone through a long process. Cellulose and its derivatives are regarded as the first generation of haemodialysis materials [3], but they are soon replaced by various of synthetic polymers due to the poor molecular specificity and aggravated condition of the patient. Several polymers including polyurethane, polysulfone (PSF) and polyethersulfone (PES) [4] represent better haemocompatibility and classy permeability and could allow the toxins with medium molecular weight to exchange. The original synthetic polymers still cause the coagulation even thrombosis when contacted with blood [5], although they partially ameliorate the permeate flux and biologic properties. Consequently, the desired biocompatible materials with anticoagulant and antibacterial features, as well as sterling mechanical and chemical stability, are still to be developed.

Among these polymers, PSF presents great potential in excellent mechanical properties, chemical stability and prominent molecular plasticity. Unfortunately, PSF lacks hydrophilicity and charged groups, which prevents the formation of hydration layers at the surface of biofilms and ulteriorly facilitates the protein adsorption resulting in blood clots [6]. Hence, modifications in hydrophilicity and charge of PSF are regarded as outstanding strategies in haemodialysis.

Generally, several methods consisting of blending, surface coating and grafting are used to improve the hydration capacity and biocompatibility of PSF membranes [7]. Among them, grafting representing outstanding chemical stability in molecular structure has become the main modification way for hydrophilic and anticoagulant factors [8]. Asatekin et al. [9] prepared the modified PSF membranes grafted by ethylene glycol and found that the great hydration capacity was attributed to the hydroxyl groups in PSF with a chlorine bridge, but the change in protein adsorption performed poorly. Xiang et al. [10] synthesized novel anticoagulant PSF membranes covalently grafted amphoteric binary copolymer with sulfonic groups through surface-initiated atom transfer radical polymerization. The resulting membranes could only act on thrombin, except to prevent exogenous coagulation. Liu et al. [11] proposed the viewpoint of 'coordination of main chain and side groups' and 'maintaining normal conformation' by the modification effects of internal salt zwitterions and explained the reasons for the changes in charge and protein adsorption, which was of great significance for the design of biological activity and electronegativity.

For further significant improvement of haemocompatibility, various natural anticoagulant factors such as heparin had been studied in depth. Huang et al. [12] demonstrated that heparin worked well in enhancing the hydration capacity and anti-protein adsorption in view of abundant sulfonic and carboxyl groups. However, adverse efforts [13–15] such as thrombocytopenia and spontaneous blending were reported to be emerged by the chronic application of heparin in clinical. Therefore, it was sunshine to hunt a similar skeleton to replace the heparin. Chitosan (CS) matched well with the great demand on structure and potential in avoiding complications and obtaining heparin-like hydration and anticoagulation [7]. In addition, abundant hydrophilic groups such as carboxyl, hydroxyl and sulfonic groups endowed CS with great hydration capacity and charges. Lima et al. [16] succeed in the introduction of sulfonic groups in CS and gained the resemble structure to heparin, debasing the protein adsorption and boosting the hydration through the interaction between modified CS with a negative charge and water molecules. Huang et al. [17] created heparin-like hydrogels (HLCSs) with the skeleton of CS by various carboxymethyl and sulfonic groups and achieved the outstanding thrombus inhibition property according to the results of the activated partial thromboplastin time and thrombin time. Furthermore, the resulting contact activation and complement activation confirmed that HLCSs possessed great blood compatibility. He et al. [18] developed the study by introducing amides and HLCSs into graphene oxide-based core @ PES-based shell beads, and the significantly prolonged activated partial thromboplastin time (APTT) and thrombin time (TT) testified that heparin-like polymer co-functionalized GO-based core @ PES-based shell beads possessed great anticoagulant property, and there might be potential in haemoperfusion for the resulting product. Medeiros Borsagli et al. [19] composed the N-acylated CS derivatives by cysteine with strong hydration capacity. The modified CS with various carboxyl groups possessed outstanding biocompatibility and anti-fouling property. In our previous studies, sulfonated hydroxypropyl chitosan (SHPCS) with hydroxyl and sulfonic groups was introduced into the pretreated PSF membranes with a glutaraldehyde bridge by Liu et al. [20]. The resulting modified PSF membranes with hydroxyl and sulfonic groups which could control the charge density and reflect great hydrophilicity, cytocompatibility and anticoagulation. But the few reaction sites of C6 in CS prevented the better biocompatibility, and neutral hydroxyl groups failed to form a synergistic effect in anticoagulation with the sulfonic groups to promote the haemocompatibility. Tu et al. [21] developed the bridge with acrylic acid and then grafted SHPCS with hydroxyl and sulfonic groups at the pretreated

PSF membranes to improve the hydrophilicity and anticoagulation. Yan *et al.* [22] optimized the modification process of chlorine bridge by 4-(chloromethyl)benzoic acid in the pretreatment of PSF membranes to suffer more active sites for SHPCS. Additionally, several attempts in our previous studies developed the exogenous coagulation pathway to elevate haemocompatibility. These attempts met the desired requirements to some extent and proved that the hydroxyl and sulfonic groups hoisted the hydration and charge capability of CS and final membranes. For the above, CS with more than two kinds of negative groups will achieve the purpose of enhancing the regulation ability to functional groups and chargeability in structure. Then, the PSF membranes adorned by the above modified CS are expected to further grow haemocompatibility.

In order to improve the biocompatibility of PSF, it seems to be an effective strategy to graft amounts of negative carboxyl and sulfonic groups into CS for the synergistic coagulation of the two kinds of groups, and modified CS was grafted into PSF. However, there are few reports on the modifications of PSF membranes by two types of negative groups to develop the anticoagulant property through the synergistic effect of these groups. In this work, sulfonated citric chitosan (SCACS) with a heparin-like structure was synthesized through N-acylation and sulfonation to take the negative carboxyl and sulfonic groups at the beginning. Then, the chloroacylated PSF (CAPSF) was prepared by introducing –COCH$_2$Cl groups into PSF to endow it with more active sites, and then the CAPSF membranes were prepared by the phase separation method. Following, the CAPSF membranes were treated by amination, and amination polysulfone (AMPSF) membranes were obtained, then SCACS was covalently bonded at the surface of AMPSF membranes, and PSF-SCACS membranes were finally obtained, which were verified by $^1$H NMR, ATR-FTIR and XPS. Furthermore, the water contact angle (WCA), porosity, SEM and a series of haemocompatible tests, including protein adsorption, platelet adhesion, haemolysis rate (HR), plasma recalcification time (PRT), APTT, prothrombin time (PT) and TT, demonstrated that the PSF-SCACS membranes pointed out that the great hydrophilicity and haemocopatibility of modified PSF membranes rose to a high level because of cooperative promotion of –COOH and –SO$_3$ groups in the formation of hydration layer and protein adsorption layer, especially in SCACS. The preparation process of PSF-SCACS membranes is shown in figure 1.

# 2. Material and methods

## 2.1. Materials

PSF with an average molecular weight of 16 000 was purchased from Aldrich Co., Ltd, China. Chloroacetic chloride (CAC, purity: greater than or equal to 98%) was obtained from Former Derivatives Technology Co., Ltd, China. Anhydrous aluminium chloride with a purity greater than 99% and CS (average viscosity: 100–200 mPa s) with deacetylation exceeding 95% were provided by Aladdin Reagent Co., Ltd, China. Dichloromethane (DCM), ethylenediamine (EDA), carbodiimide (DCC), hydrochloric acid, 25% of glutaraldehyde solution, citric acid (CA), acetic acid and formamide were all supplied by Sinopharm Chemical Reagent Co., Ltd, China. Chlorosulfonic acid was bought by Mars Fine Chemicals Co., Ltd, China. N, N-dimethylacetamide (DMAC) was used as the solvent to cast a film, and absolute ethanol was obtained from Fuyu Fine Chemical Co., Ltd, China. MD44 dialysis bags (average diameter: 28 mm) with the retention range from 8000 to 12 000 were bought from Leibusi company, China. All chemical reagents in this experiment were analytical pure.

## 2.2. Preparation of polysulfone-modified membranes

The process of modified membranes includes the following steps, which are a synthesis of SCACS, pretreatment of PSF membranes (preparation of CAPSF by chloroacylation of PSF, preparation of CAPSF membranes by phase separation method and preparation of AMPSF membranes by amination of CAPSF membranes), preparation of PSF-SCACS membranes by grafting SCACS on the surface of AMPSF membranes.

### 2.2.1. Synthesis of sulfonated citric chitosan

For weakening the intermolecular force of CS, 20 g of CS was added into a 250 ml breaker with 80 g of sodium hydroxide and 150 ml of ultrapure water under stirring at room temperature. After mixing uniformly, the mixture was frozen in a refrigerator for 10 days. The alkalized CS was obtained after thawing.

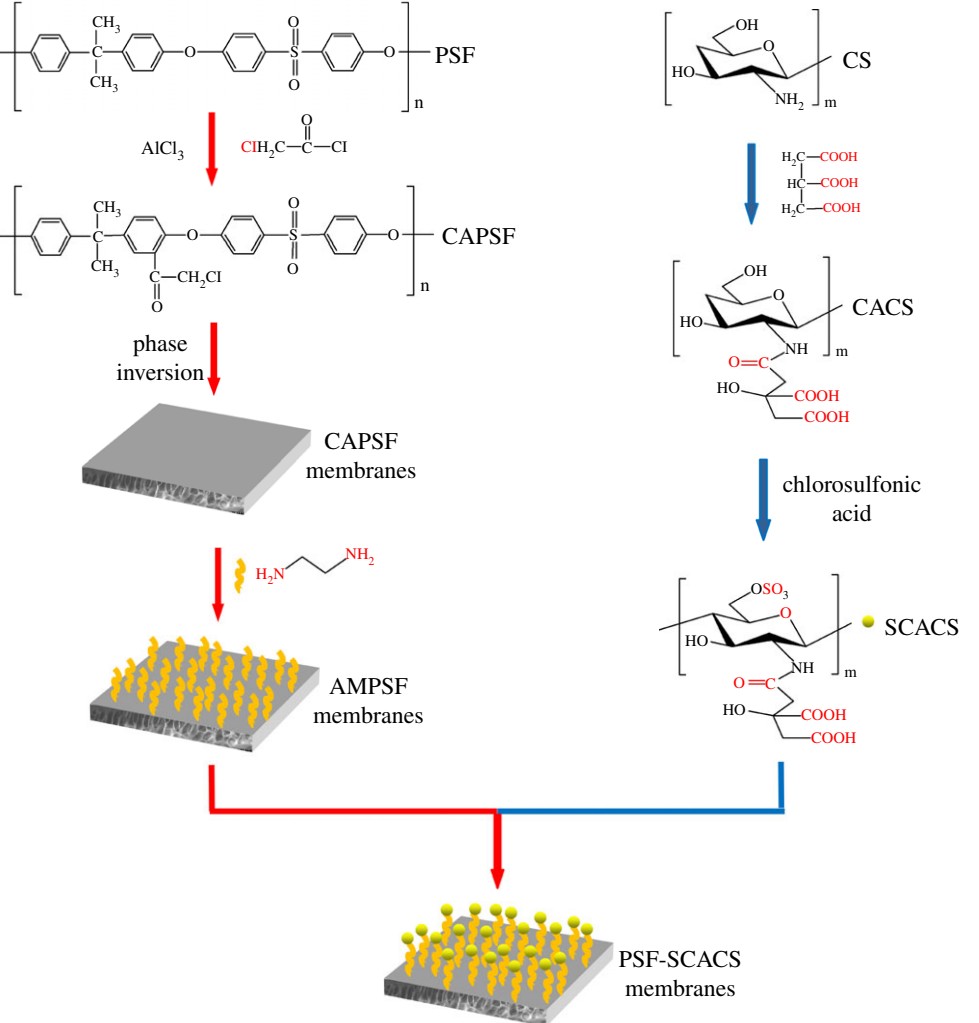

**Figure 1.** The preparation process of PSF-SCACS membranes.

Afterwards, 12 g of CA dissolved in 90 ml of deionized water and alkalized CS (10 g) were mixed in a three-port flask under stirring, followed by reflux at 60°C for 3 h. Then, another 100 ml of deionized water was slowly added into the solution under stirring after the reaction. Next, the pH of the solution was adjusted to neutral with an appropriate amount of hydrochloric acid, and the product was precipitated in acetone. Citrate chitosan (CACS) was finally obtained by washing with ethanol three times.

In order to prepare a series of different sulfonation reagents, 4 ml of CA was gradually dropped into 20 ml of formamide in the ice region below 5°C. Following, 2 g of CACS was added into a 150 ml round-bottomed flask along with sulfonation reagents, and the system was stirred to reflux at 60°C for a certain time (six groups: 1, 2, 3, 4, 5 and 6 h). Then, 100 ml of ultrapure water was put in the system, followed by transferring the solution to a dialysis bag for removing small soluble impurities at room temperature. After 24 h, the pH of dialysate was adjusted to 12 with sodium hydroxide solution (2 mol l$^{-1}$), and then the pH of solution was regulated to 3 with hydrochloric acid (3 mol l$^{-1}$), set for 20 min, dialysis for another 2 days. Finally, SCACS was acquired by concentrating the dialysate in a vacuum drying cabinet at 50°C. In addition, the SCACS-1 indicated that the synthesis reaction time of 2 g of CACS with 4 ml of chlorosulfonic acid was 1 h, so did other SCACS series.

## 2.2.2. Pretreatment of polysulfone membranes

With the purpose of increase of reactive groups in PSF, chloroacetylation was adopted in the pretreatment of PSF membranes at the beginning by Qiao *et al.* [23]. First, 2 g of PSF was dissolved into 40 ml of DCM in a 150 ml three-necked flask. Next, 0.8 ml of CAC (10.1 mmol) as a chloroacyl reagent was stirred to

dissolve in this system for 10 min below 5°C. After that, 0.9 g of aluminium chloride catalyst was added into the flask in three portions. Following the reaction, the system was placed under nitrogen protection to ensure the response to the target, and the Friedel–Crafts reaction between PSF and chloroacyl reagent was controlled a certain time at 10°C (six groups: 1, 3, 6, 9, 12 and 15 h), while the variable groups of temperature were implemented at 5°C, 10°C, 15°C, 20°C, 25°C and 30°C. Then, the coarse product was separated with ethanol and a few drops of hydrochloric acid for removing the unreacted aluminium chloride. As a result, CAPSF was as a produce after washing, filtering and drying at 50°C. The grafting rate of chloroactyl groups could be calculated by the following:

$$\text{grafting rate} = \frac{3A_2}{A_1} \times 100\%,$$

where $A_2$ was the integral area of proton hydrogen in –$COCH_2Cl$ at 4.62 ppm in [1]H NMR, and $A_1$ was the integral area of proton hydrogen in –$CH_3$ at 1.56 ppm. All values were the average of three independent experiments.

The CAPSF/PSF membranes maintained the mechanical behaviours in modification of PSF, which were similarly obtained from Liu *et al.* [20]. The CAPSF membranes were prepared by phase separation method, and the casting CAPSF solution consisted of 8% CAPSF, 10% PSF and solvent DMAC. All CAPSF membrane samples were cut into 2 × 2 cm.

Sixty pieces of CAPSF membranes (2 × 2 cm) were completely immersed in 30 ml of EDA [12] for 20 min at 25°C for easier access between PSF membranes and SCACS, and the coarse AMPSF membranes were obtained. After that, the AMPSF membranes were rinsed three times with deionized water and then dried at 30°C. All AMPSF membranes were cut into 1 × 1 cm.

### 2.2.3. Preparation of polysulfone-sulfonated citric chitosan membranes

In view of the difficulty of amide reaction between polymers, the activator was necessary. For this, 3 mol l$^{-1}$ of DCC buffer was based on the acetic acid solution to enhance the activity of carboxyl groups. The DCC buffer was based on the acetic acid solution to enhance the activity of carboxyl groups. One gram of SCACS was put into a 150 ml three-necked flask with 20 ml DCC solution (60 mmol) and activated at 4°C for 3 h under stirring. Immediately, 15 pieces of AMPSF membranes (1 × 1 cm) were fully infiltrated in the solution at the same temperature for a certain time (3, 6, 9, 12, 15 and 24 h). Afterwards, the preliminary results were reacted with the 2 mmol l$^{-1}$ of glutaraldehyde solution at 25°C for 1 h. Then, the resulting membranes were completely rinsed with phosphate buffer and deionized water to get rid of excess glutaraldehyde, and dried at 50°C for 24 h. The PSF-SCACS-3 represented the PSF membranes modified by SCACS with a reaction time of 3 h, as did the others. The grafting rate could be calculated by the following formula:

$$\text{grafting rate} = \frac{M_1 - M_0}{M_0} \times 100\%,$$

where $M_1$ (mg) and $M_0$ (mg) were the mass of PSF-SCACS membranes and AMPSF membranes, respectively. All values were the average of three independent experiments.

### 2.3. Characterization

The functional groups between the raw materials and products were characterized by FTIR (Nicolet iS50, USA), and the chemical compositions in original and modified PSF membranes were all analysed by an ESCALAB 250Xi XPS instrument. A Flash 2000 affirmed the element analysis (EA) of CS, CACS and SCACS. The synthesized CAPSF was tested by [1]H NMR with a 500 MHz BRUKER spectrometer.

### 2.4. Morphology and hydration capacity

The morphology of the membranes was observed by SEM on a JSM-7610F field emission scanning electron microscope. The resulting membranes were placed at 1000×, 10 000× and 1500× at a voltage of 15 kV, respectively, and the distribution range of surface element.

The hydration capacity of pristine and modified PSF diaphragms depended on the WCA which was measured by an SL 200B contact angle goniometer (KINO, USA). In addition, the internal structure effected partially the biocompatible performances, and the porosity of membranes was

applied to evaluate the internal hydrophilicity [24,25]. The porosity was calculated by the following equation:

$$\text{porosity} = \frac{M_1 - M_0}{Al\rho} \times 100\%,$$

where $M_1$ (g) and $M_0$ (g) were the quality of pure and modified membranes, respectively, $A$ (m$^2$) was the effective area of surface, $l$ (m) represented the thickness of diaphragms and $\rho$ was the density of water (998 kg m$^{-2}$). All results were the average of three tests.

## 2.5. Haemocompatibility

The haemocompatibility tests were supported by the Xiangya Medical College, Central South University, China. The blood samples were from a normal 25-year-old female and in compliance with relevant laws and regulations, and were used for all the following tests including platelet adhesion and deformation, HR, plasma recalcation time, APTT, PT and TT. In tests, the results were the averaged value in three independent parallel experiments. The credibility and accuracy of measurements were pledged by the professionalism and rich experiences of operators. The blood samples were 168 in this study.

### 2.5.1. Protein adsorption

Protein adsorption as the prime factor leading to the formation of thrombus was measured by the bovine serum protein (BSA) adsorption test [26]. Based on preparation works, the desired 1 mg ml$^{-1}$ of protein adsorbent was made up of 0.1 g of BSA and the phosphate buffer solution (PBS) at pH of 7.4. At first, the PSF, AMPSF and PSF-SCACS membranes were separately immersed in the PBS for 24 h and in the 1 mg ml$^{-1}$ of protein adsorbent at 37°C for another 2 h. After washing with PBS, all membranes were transferred into 20 ml of SDS (2 wt%) detergent under stirring slightly at 37°C for 2 h so as to elute the adsorbed protein into SDS solution. Afterwards, the adsorption capacity of the simulated contact was subjected to a UV-1801 spectrophotometer at 280 nm. All results were the average of three tests. The blood samples were from a normal 25-year-old female and in compliance with relevant laws and regulations.

### 2.5.2. Platelet adhesion and deformation

Platelet adhesion and deformation, as the biological phenomenon of contact behaviours between the blood and materials, were carried out in *in vitro* simulated blood environment [20]. Primarily, the platelet-rich plasma (PRP) was removed from the upper layer in an anticoagulant collection tube with a sample for 10 min at the centrifugal speed of 1000 r.p.m. Then, the pure and modified PSF membranes (1 × 1 cm) were immersed in the centrifuge tubes with normal saline at 37°C for 1 h. Next, each centrifuge tube poured out the saline, was injected 1 ml of PRP and stirred at 37°C for 2 h. The treated films were washed thrice by saline, followed by immersing in 2.5 wt% glutaraldehyde saline at 4°C to fix platelets. After 24 h, the films were cleaned again and soaked in 25%, 50%, 75% and 100% alcohol-water solution for 15 min, respectively. Finally, the diaphragms in test were dried and sprinkled with gold. The number and morphology of platelets adhered to the membranes were recorded by SEM at 1000×, 10 000× and 30 000×.

### 2.5.3. Haemolysis rate

HR represented the possibility of decomposition and dissolution of red blood cells [20], and the measurement method was as follows. First, 4 ml of whole blood was dissolved in 24 ml of saline as a backup dilute solution and then implanted separately 3 ml into each centrifuge tube. Meanwhile, 0.5 ml of fresh blood was diluted with 3 ml of ultrapure water as a positive control group, and the corresponding dilute solution with 0.5 ml of blood and 3 ml of saline could be as a negative one. After warming up at 37°C for 1 h, the samples (PSF and all modified PSF membranes) were separately placed in centrifuge tubes at 37°C for another 3 h and followed centrifuging at 500 r.p.m. for 10 min. As a result, the supernatant was the detection liquid on the target, which was measured and recorded with an ultraviolet spectrophotometer at 545 nm. HR was calculated by the following:

$$\text{haemolysis rate} = \frac{D_t - D_{nc}}{D_{pc} - D_{nc}} \times 100\%,$$

where $D_t$ represented the absorbance value of sample supernatant, while $D_{nc}$ and $D_{pc}$ represented individually the absorbance value of supernatant in the negative and positive control group. All results were the average of three tests.

### 2.5.4. Plasma recalcation time

Plasma recalcation time was a test on endogenous coagulation, playing a significant role on anticoagulant [21]. For platelet-poor plasma (PPP), the collection of fresh blood was placed in a centrifuge at 3000 r.p.m. for 15 min, and the upper layer was the required solution. The pristine and modified PSF membranes (1 × 1 cm) were immersed in PBS solution at 37°C for 1 h, and then they were added in a centrifuge tube with 0.5 ml of PPP and 0.5 ml of calcium chloride solution (0.025 mol l$^{-1}$) under a 37°C water bath. The time at the first appearance of insoluble flocculent fibrin was recorded as PRT. All results were the average of three tests.

### 2.5.5. Activated partial thromboplastin time, prothrombin time and thrombin time

APTT, PT and TT were determined after simple processing of membrane samples. Primarily, PSF and PSF-SCACS membranes (1 × 1 cm) were soaked into PBS solution at 40°C for 1 h. Then, the membrane samples were carried in the centrifuge tubes containing 1 ml of PPP, and heated at 37°C for 1 h. Finally, the membrane samples were sent into an automatic coagulator (CA-7000) for APTT, PT and TT. In view of the small difference, we conducted a statistical analysis by SPSS 16.0 software for groups. Pearson's correlation analysis was performed. $p < 0.05$ was considered to indicate statistical significance.

## 3. Results and discussion

### 3.1. Preparation and characterization of polysulfone-sulfonated citric chitosan membranes

#### 3.1.1. Synthesis of sulfonated citric chitosan

The successful synthesis and sulfonation rate were crucial for the activity of SCACS. The resulting spectrum is shown in figure 2b, the peaks between 3100 and 3500 cm$^{-1}$ representing the stretching vibrations of –NH$_2$ and –OH occurred with strong absorption when compared with CS and CACS, which were attributed to the reaction of amino and carboxyl groups. And then a new characteristic peak of –COOH appeared in CACS at 1710 cm$^{-1}$, while a peak in CS was migrated at 1594 cm$^{-1}$ in CACS owing to the vibration of amide III band. Furthermore, the peak at 1468 cm$^{-1}$ emerged a sign of weakening, indicating that the reaction sites were on the amino groups in CS. Above all, CACS was synthesized. For SCACS, the peak at 3169 cm$^{-1}$ had a strong stretching vibration, which was traceable in the reaction of sulfonic and hydroxyl groups. Meanwhile, the peak at 1594 cm$^{-1}$ in SCACS appeared a stretch signal, which might suggest that unexpected sulfonic groups were unexpectedly bound with some free amino groups. Another fact was that the peak at 1468 cm$^{-1}$ became slightly stronger in SCACS, which was owing to the C=O of –NHCOCH$_3$ after sulfonation. Additionally, the new peak at 1246 cm$^{-1}$ belonging to S=O of –SO$_3$ and the peak of ether bond in C-O-S at 806 cm$^{-1}$ pointed out that sulfonic groups were introduced into CACS. Consequently, it appeared that the modification from CS to CACS and SCACS was progressed smoothly and successfully.

For sulfonation rate, it could be regarded as the ratio between the carboxyl and sulfonic groups for further research. According to figure 2a, the colour of the products had gradually deepened from light yellow to bright yellow to dark brown with the reaction time being extended, which testified that the degree of sulfonation was deepened by degrees. In other words, it qualitatively proved the above fact that SCACS was successfully synthesized. According to the results of EA in table 1, the content of S in SCACS samples became higher than the previous one from 7.90% to 12.87%, while the content of C decreased correspondingly from 26.45% to 24.08%. The change of the two elements was attributed to the increase of sulfonic groups in SCACS which was affected by the reaction time. With comprehensive consideration of S content in natural heparin (8.89%–12.67%) [22], the optimum sulfonation time was 5 h.

#### 3.1.2. Pretreatment of polysulfone membranes

The pretreatment for PSF could provide a strong support for further modification. Chloroacetylation suffered PSF with chlorine reactive sites to promote the reactivity and pledged the smooth progress of the next step. Therefore, the $^1$H NMR spectra and the grafting rates of chloroacetyl groups are

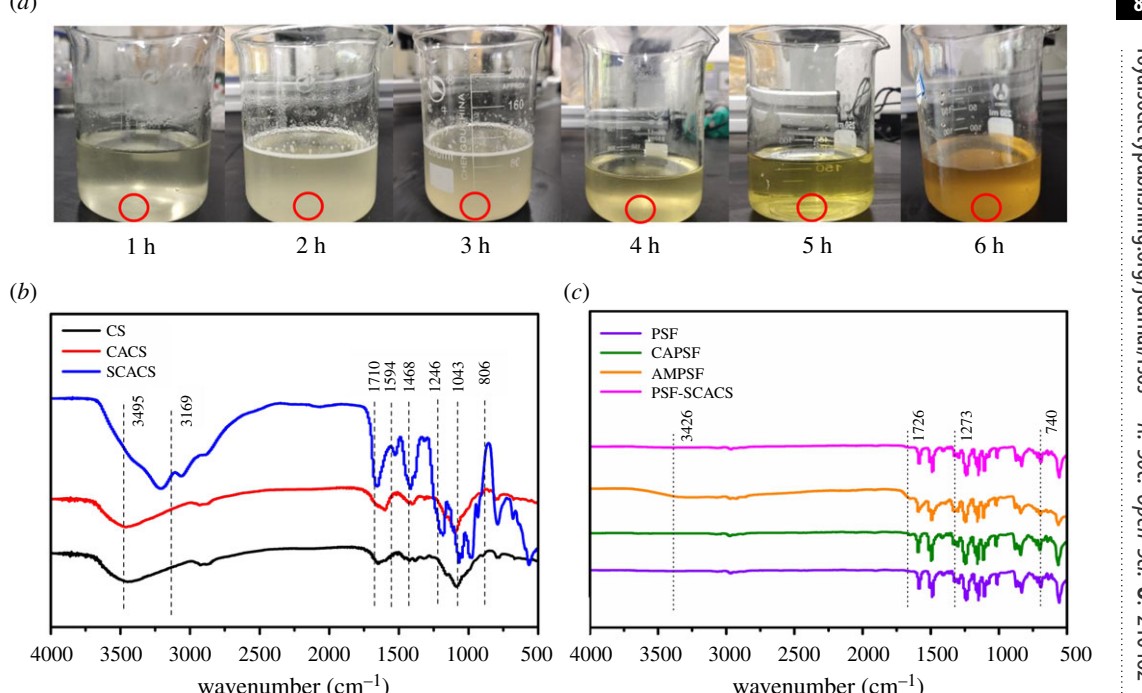

**Figure 2.** (a) Colour changes with the prolonged reaction time in sulfonation. FTIR spectra of (b) CS, CACS and SCACS and (c) PSF, CAPSF, AMPSF and PSF-SCACS membranes.

**Table 1.** The element composition of CACS and SCACS by element analysis.

| membranes | element content (%) | | | |
|---|---|---|---|---|
| | C (%) | H (%) | N (%) | S (%) |
| CACS | 25.75942 | 3.86475 | 2.67681 | 0.00000 |
| SCACS-1 | 26.08752 | 4.12897 | 3.05143 | 7.89851 |
| SCACS-2 | 26.34586 | 3.79431 | 2.98514 | 9.36592 |
| SCACS-3 | 25.89721 | 3.98641 | 3.01469 | 10.54687 |
| SCACS-4 | 24.15689 | 3.57620 | 3.16987 | 12.15676 |
| SCACS-5 | 24.26594 | 3.96853 | 3.29643 | 12.46948 |
| SCACS-6 | 24.07634 | 4.10265 | 3.03594 | 12.86517 |

presented in figure 3. Compared with the spectrum of PSF, there occurred a new peak at 4.62 ppm, which was the signal of H in –CH$_2$Cl and demonstrated the smooth process of chloroacylation. In figure 3b, the grafting rates of chloroacetyl were in a growing line but finally levelled off with the increase of reaction time. The above phenomenon indicated that the reaction was controlled by kinetics and would gradually move to the positive direction until the reactants were exhausted. As a result, the highest grafting rate reached 23.27% when reacted for 15 h. But taking the consumption of raw materials into consideration, 9 h was regarded as the optimum reaction time. For temperature, the reaction would go smoothly by dynamics, and the grafting rate also rapidly grew at 5°C to 15°C while the growth trend would slow down at 20°C to 30°C. In experiments, chloroacetylation pertained to an exothermic reaction, which would lead to an amount of cross-linking at high temperature. Meanwhile, aluminium chloride was easily decomposed at a higher temperature. For the above, the suitable reaction temperature was 15°C. According to the study by Huang *et al.* [12], the grafting density of amino groups was determined by golden orange stain as 8.98 mmol cm$^{-2}$. The successful modification of chloroacetylation and amination were proved in figure 4. It was shown that the content of C decreased and the content of O increased after grafting –CH$_2$COCl, which demonstrated that the CAPSF membranes were successfully synthesized, and a new peak of nitrogen of AMPSF membranes indicated that amino groups were smoothly introduced into CAPSF membranes.

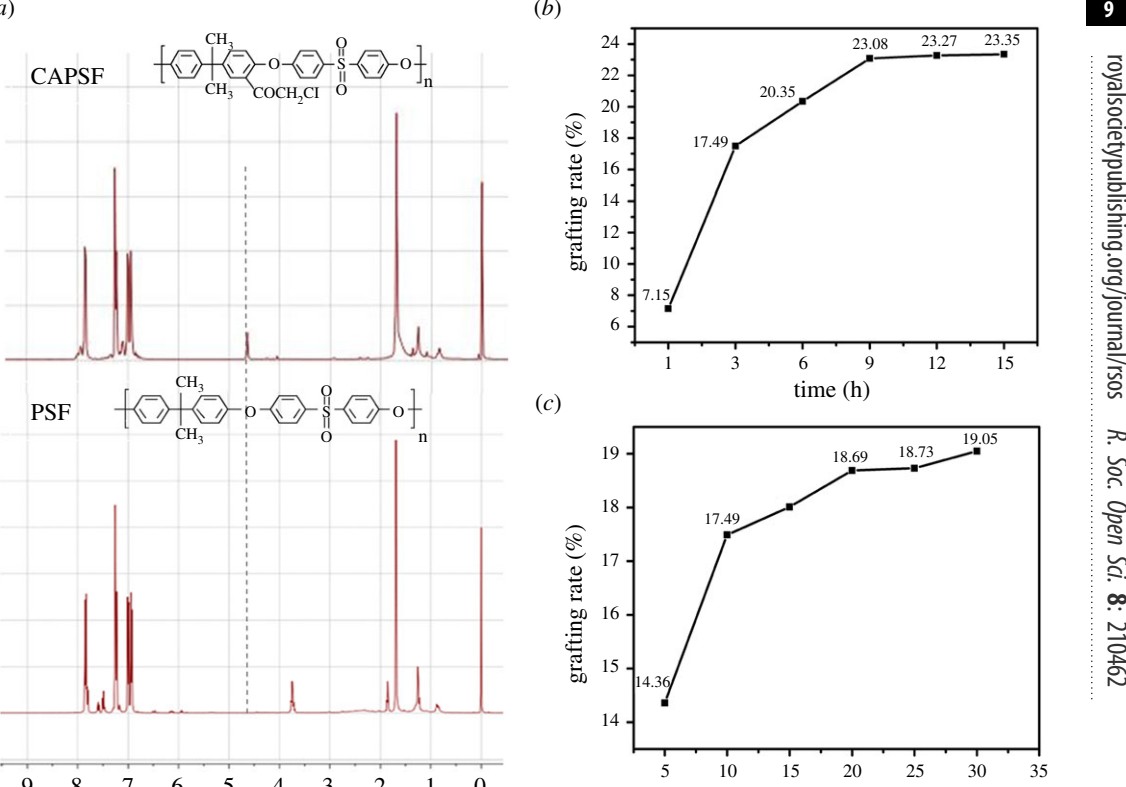

**Figure 3.** (*a*) $^1$H NMR spectra of CAPSF and PSF. Different effects of (*b*) treatment time on grafting rates of −Cl (at 10℃) and (*c*) reaction temperature on grafting rates of −Cl (at 3 h). All data were calculated by the mean value of three independent measurements.

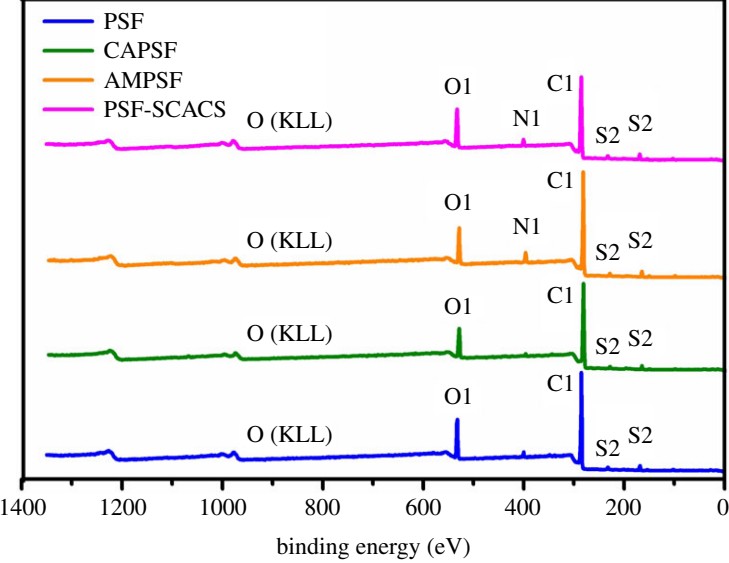

**Figure 4.** XPS spectra of PSF, CAPSF, AMPSF and PSF-SCACS membranes.

### 3.1.3. Preparation of the polysulfone-sulfonated citric chitosan membranes

For making an acquaintance for further modification, the groups at the surface of original and modified PSF membranes were tracked by FTIR, and the spectra are shown in figure 2*c*. Although the change in characteristic peaks was obscure due to the low grafting density of the amino groups, the peak of signal of −NH$_2$ appeared at 3426 cm$^{-1}$ by local magnification. Furthermore, the characteristic peak of C=O at

**Table 2.** The elemental composition of pristine PSF membrane and modified membranes at the surface by XPS.

| membranes | element content (%) | | | | |
|---|---|---|---|---|---|
| | C(%) | O(%) | N(%) | S(%) | Cl(%) |
| PSF | 81.34 | 16.42 | — | 2.24 | — |
| CAPSF | 76.97 | 18.65 | — | 2.17 | 2.21 |
| AMPSF | 78.03 | 13.91 | 5.51 | 2.42 | 0.13 |
| PSF-SCACS | 68.16 | 23.96 | 4.84 | 2.98 | 0.06 |

$1726\ cm^{-1}$ appeared stretching vibration, which was owing to the introduction of –COCH$_2$Cl in PSF. In addition, the peak at $740\ cm^{-1}$ was the vibration of C-H of the –COCH$_2$Cl. Otherwise, the peak at $1273\ cm^{-1}$ in PSF-SCACS membranes was weaker than in PSF and AMPSF membranes, which was associated with the signal of S=O. These results achieved the desired targets and had successfully justified the great preparation of CAPSF, AMPSF and PSF-SCACS membranes.

For further verification of the above results, pristine and modified PSF membranes were examined by XPS, and the spectra are displayed in figure 4. In table 2, the samples between PSF and AMPSF membranes emerged a great difference in the content of O and C, and without N in PSF. The content of O increased significantly from 16.42% to 23.96% before and after PSF membranes were modified by SCACS; by contrast, the carbon content decreased from 81.34% to 68.16%, the content of N and S of AMPSF membranes were 5.51% and 2.42%. The content of N and S of PSF-SCACS membranes became 4.84% and 2.98%. In summary, all explanations represented the successful synthesis of AMPSF and PSF-SCACS membranes.

According to figure 10d, the grafting rates of SCACS in PSF-SCACS membranes occurred by a growth trend in pace with the growth of reaction time, which presented the rapid ascent before 15 h but put a drag on growth from 15 h to 24 h. At the same time, the highest grafting rate reached 12.16% at 24 h while achieved 12.09% at 15 h. Accordingly, 15 h was considered as the optimal reaction time of SCACS.

### 3.1.4. Morphology and hydration capacity of polysulfone-sulfonated citric chitosan membranes

Great mechanical properties and quality permeability endowed PSF with great potential in haemodialysis, which suggested the momentous role of membrane structures in maintaining these properties. Therefore, the morphologies of both original and modified PSF membranes were observed under SEM and recorded in figure 5. It was evident that each image showed an asymmetric porous structure at a magnification of 1500×, but the finger-like structures were extended after grafting SCACS. In addition, the surface became rough after the modification. At the same time, the distributions of S, O and C in AMPSF and PSF-SCACS membranes (product reacted for 15 h) were uniform by EDS (in figure 6). In other words, the target products had been successfully synthesized, and the finger-like structure of PSF was a slight change during modification.

For further exploration of structures of membranes before and after modification, the porosity was detected to assess the extent of expansion which is shown as figure 7a. A satisfied fact certificated the low degree of expansion with 22.65%, while the porosities of pristine and final membranes were 14.73% and 38.24%, which supported the discussion relevant to internal membrane structures on light damage after grafting SCACS by SEM.

In view of the hydration capacity of the surface, WCA is a simple way to estimate the hydration capacity. The WCA results of all membranes are shown in figure 7b. Compared with pristine membranes, the WCA of AMPSF membranes dropped 6°, demonstrating a slight improvement of hydrophilicity. After grafting SCACS, the WCA of modified PSF membranes with heparin-like structure reduced to 54° and abated by degrees with the enhancement of reaction time. In addition, the lowest WCA, that of the PSF-SCACS-24 membrane, was 39°. In a conclusion, the hydration capacity at the surface of PSF-SCACS membranes had improved constantly with the prolonged grafting time.

## 3.2. Haemocompatibility of modified polysulfone membranes

### 3.2.1. Protein adsorption

Protein adsorption, the first step of cruor according to the coagulation mechanism, is regarded as the primary indicator of evaluating blood compatibility [27]. To approach the real physiological conditions,

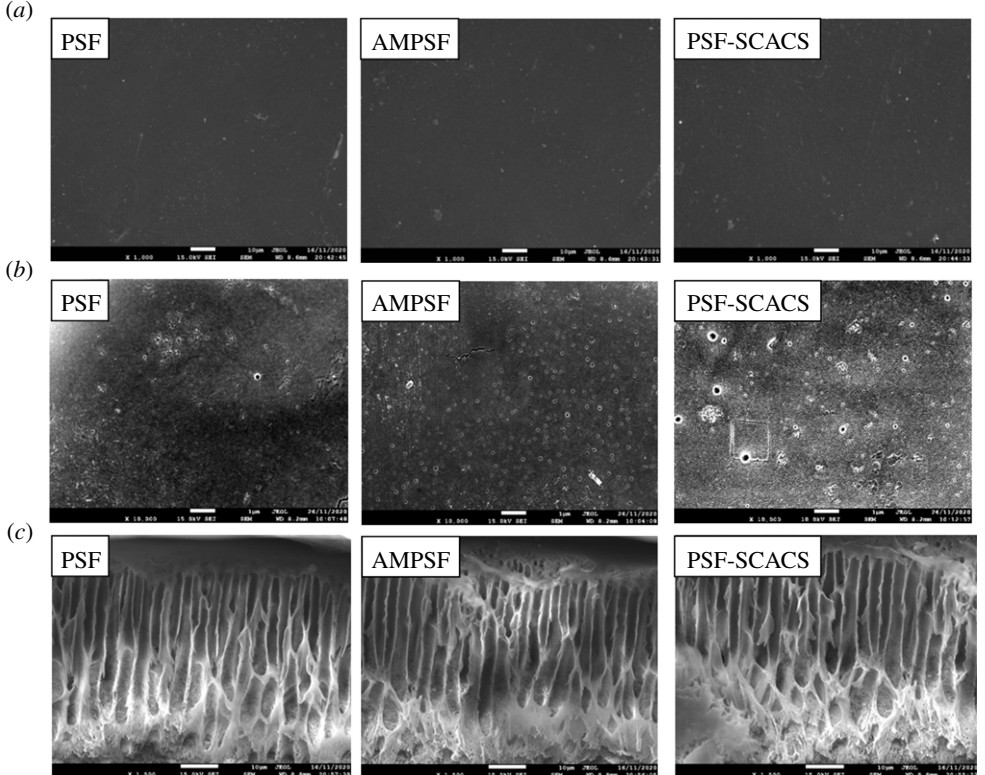

**Figure 5.** SEM images of PSF, AMPSF and PSF-SCACS membranes. Surface morphologies (*a*) at a magnification of 1000× and (*b*) at a magnification of 10 000×; (*c*) internal morphologies at a magnification of 1500×.

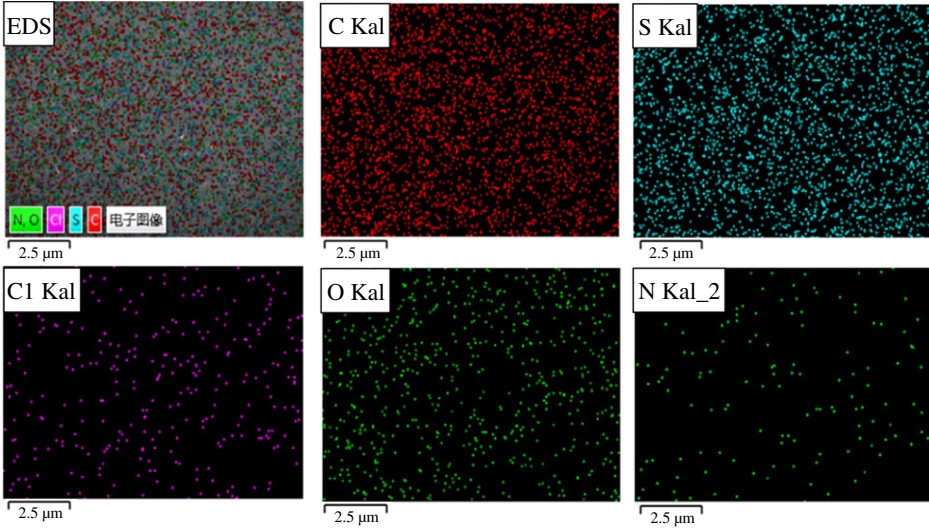

**Figure 6.** EDS images at the surface of PSF-SCACS membranes.

the BSA is adopted to simulate human serum albumin in virtue of the similar molecular weight and Stokes radius, and the results in modification are shown in figure 8. The amount of protein adsorption at the surface of PSF membranes was 362 µg cm$^{-2}$ and the amount for AMPSF membranes decreased at 345 µg cm$^{-2}$, which suggested that the amination had little effect on anti-protein adsorption. However, the adsorption amount of BSA was obtained as 135 µg cm$^{-2}$ after grafting SCACS, almost one-third of that of PSF membranes. Concurrently, the adsorption value of BSA had reduced from 186 to 135 µg cm$^{-2}$ as the grafting time expanded, which kept the same discipline as aforesaid WCA results. Thus, these conclusions could be drawn from the above results that the promoted hydrophilicity at the surface of membranes reduced the amount of protein adsorption, and the PSF-SCACS membranes possessed

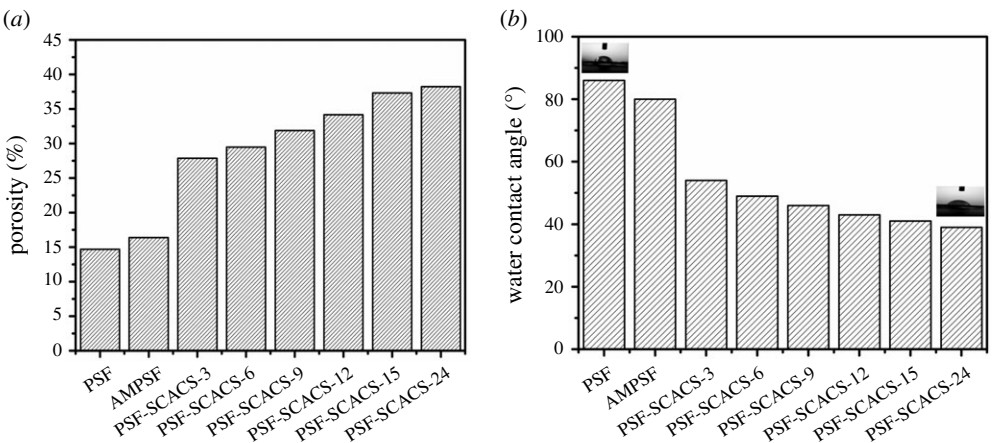

**Figure 7.** (*a*) Porosities and (*b*) WCA of pristine and modified membranes. Data were expressed as the mean value of three independent experiments.

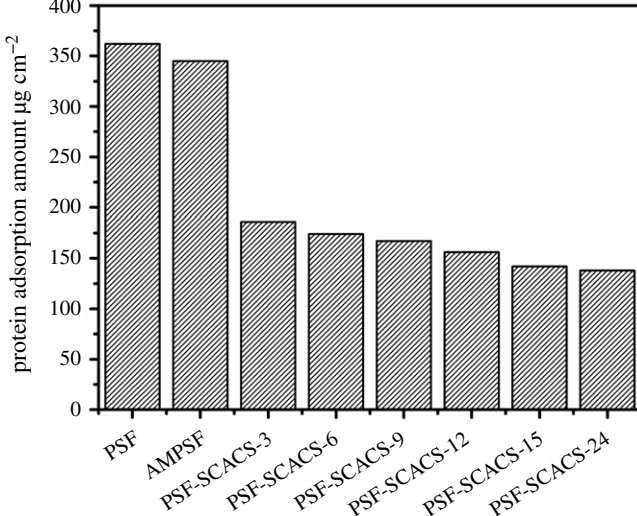

**Figure 8.** BSA adsorption at the surface of PSF, AMPSF and PSF-SCACS membranes. Data were expressed as the mean value of three independent experiments.

sterling ability to resist protein adsorption. Noticeably, SCACS had a great performance in elevating the protein adsorption resistance of PSF membranes.

### 3.2.2. Platelet adhesion and deformation

Platelet adhesion would occur after the formation of protein adsorption layer, and the shapes of platelets would change from a circle to a flat, irregular until they grew the antennae-like pseudopod [28]. Ultimately, the thrombus is caused by the activated clotting factors after aggregation and deformation of platelets. Thus, the platelet adhesion and deformation are suitable to evaluate the haemocompatibility at the material surfaces. The images of platelet adhesion at the surfaces of pure and modified PSF membranes are revealed in figure 9. By figure 9, there were a great quantity of platelets with prominent irregular appearance adhered at the surface of PSF and AMPSF membranes. Nevertheless, the amount of platelets at the surface of PSF-SCACS-15 membranes had been significantly decreased. Meanwhile, the platelets almost maintained normal shapes. The above results were consistent with BSA, which authenticated that PSF-SCACS membranes restrained effectively the platelet adhesion and deformation for further exaltation on haemocompatibility.

### 3.2.3. Haemolysis rate

HR, a way to directly cause thrombus, is defined as the release of haemoglobin into plasma after the damage of red blood cells [27], which is considered as one of the key indicators in the haemocompatibility test.

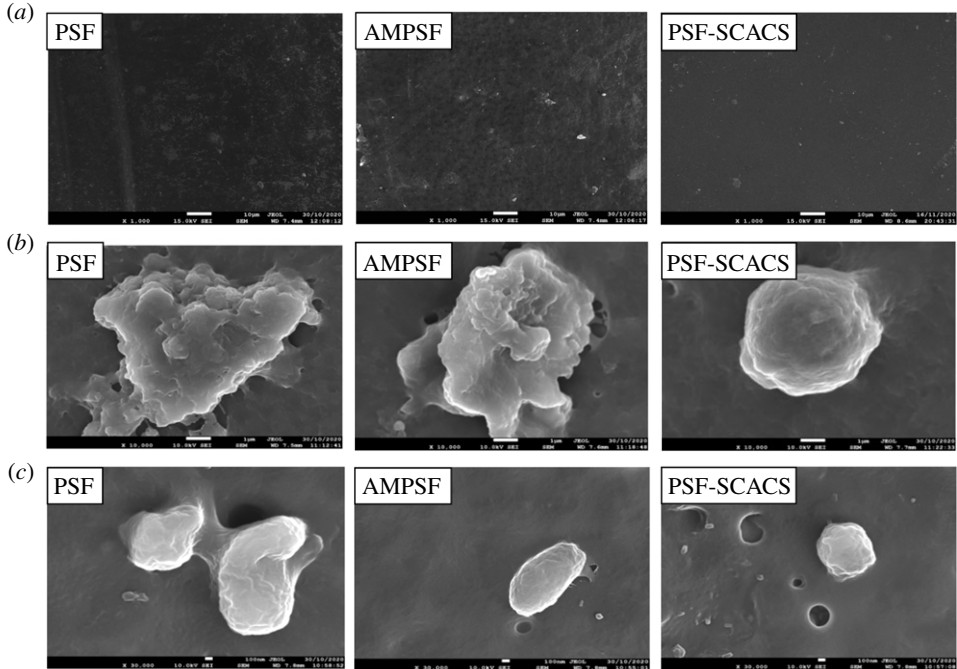

**Figure 9.** Platelet adhesion at the surface of PSF, AMPSF and PSF-SCACS membranes at (*a*) magnification of 1000×, (*b*) magnification of 10 000× and (*c*) magnification of 30 000×.

The results of HR between pristine and modified PSF membranes are presented in figure 10*a*. The HR of PSF membranes was 3.46%, while the HR of AMPSF membranes was 4.69% with a possible reason on acceleration for the rupture of red blood cells by the interaction between the erythrocyte membrane and positive charge of amino groups. Then, the HR decreased by less than 1.6% after the modification of PSF membranes grafted by SCACS. In addition, the change of HR retained a similar trend compared with other haemocompatible tests, and it might be a result that a protective layer outside the erythrocyte membrane was formed by the increased hydration after grafting SCACS. All in all, the final HR in PSF-SCACS membranes was far below the international standard of ISO 10993, and the PSF-SCACS membranes possessed brilliant cytocompatibility.

### 3.2.4. Plasma recalcification time

In endogenous coagulation, the prothrombin is converted as thrombin through the interaction between a sequence of coagulation factors in blood after the appearance of activated coagulation factor XII [29]. Then, the fibrinogen is transformed to a single fibrin until the final insoluble cross-linked fibrin with the assistance of $Ca^{2+}$. PRT, a nearly realistic simulation method for endogenous coagulation, is defined as the time required for calcium-free plasma with calcium. In figure 10*b*, the PRT of the PSF membrane was 154 s, and it had hardly changed as 159 s at the surface of AMPSF membranes. But for PSF-SCACS membranes with a reaction time of 3 h, the PRT (165 s) was prolonged a little, while it was increased to 189 s in PSF-SCACS-15 membranes. It was worth noting that the change of PRT distinctly occurred with the grafting time over 15 h. In a word, the anticoagulation had been markedly elevated at the surface of PSF membranes after grafting SCACS.

### 3.2.5. Activated partial thromboplastin time, prothrombin time and thrombin time

According to reports, APTT, PT and TT are widely applied in the preliminary detection of coagulation and screening of new drugs, as well as anticoagulant evaluation of biomaterials *in vitro* [20,30,31]. APTT is a momentous index for endogenous coagulation screening [30,31], and PT is a general indicator of exogenous coagulation pathway [20], while TT stands for extracorporeal plasma thrombin time. All membranes and SCACS were analysed by APTT, PT and TT, and the results are presented in figure 10*c*. For SCACS, the results of APTT (186.0 s), PT (33.8 s) and TT (62.6 s) demonstrated that SCACS performed perfectly both in endogenous and exogenous pathways. For APTT, the change between PSF and AMPSF membranes was less than 6 s, but the value of PSF-SCACS membranes

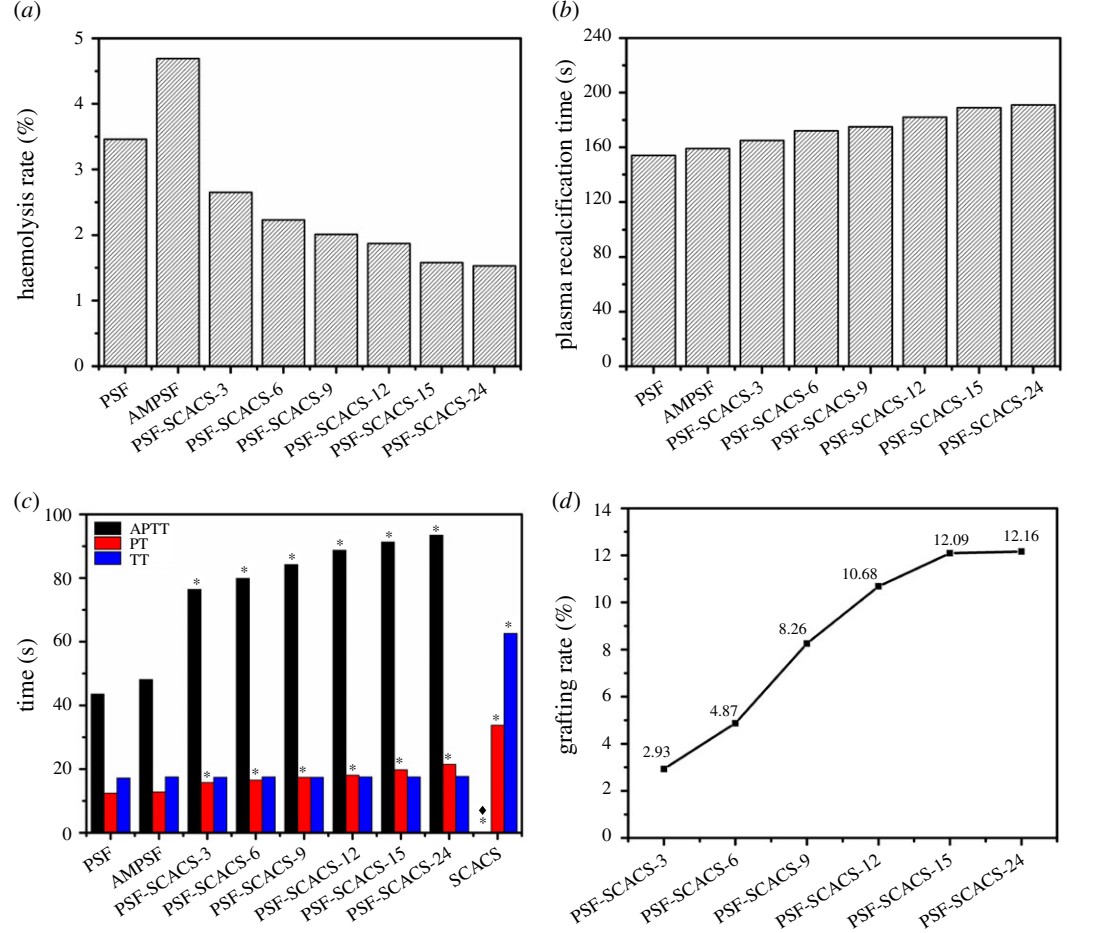

**Figure 10.** (*a*) Haemodialysis rates of pristine and modified PSF membranes. (*b*) PRTs of pristine and modified PSF membranes. (*c*) APTT, PT and TT of pristine and modified PSF membranes. (Diamond mean APTT (186 s) of SCACS, $^*p < 0.05$ compared with pristine PSF membrane.) (*d*) Effects of treatment time on grafting rates of SCACS. Data were expressed as the mean value of three independent experiments.

became more than 90 s ($p < 0.05$) which was almost twice as much in PSF membrane. With consideration of PT and TT, the values of PSF were 12.4 and 17.2 s, respectively, and they turned to 12.8 and 17.5 s in AMPSF membranes. It was an unsatisfied fact that the PT of PSF-SCACS membranes was twice that of the PSF while TT increased only 0.5 s, which indicated that TT as an index was not statistically significant ($p > 0.05$). As a result, a range of facts clearly demonstrated that PSF-SCACS membranes possessed palmary anticoagulant, and the change of APTT was much higher than TT and PT due to the amelioration of endogenous coagulation pathway.

# 4. Conclusion

The modified PSF-SCACS membranes were successfully prepared and characterized by FTIR, [1]H NMR and XPS. During the modification, the distinctive finger-like structure of membranes almost changed little. Simultaneously, the obtained PSF-SCACS membranes were homogeneous and all elements on the membranes were distributed in a symmetrical range by mapping. The resulting membranes possessed outstanding hydration capacity which greatly diminish the amount of protein adsorption. In addition, the PSF-SCACS membranes performed distinguished haemocompatibility consisting of higher anti-protein adsorption, lower platelet adhesion and deformation, reduced HR, extended PRT, prolonged APTT, PT and TT with comparison for PSF membranes. All representation analysis had proved that the modified PSF-SCACS membranes had good anticoagulation and biocompatibility, and SCACS played an important role in anticoagulation. Otherwise, PSF-SCACS membranes inhibited the thrombosis through endogenous coagulation pathway. Furthermore, there are still some works to be

studied in the future, such as the solute clearance and the amounts of platelet adhesion of pristine and modified membranes.

Ethics. All blood tests fitted the appropriate ethical approval and licences, and complied with relevant legal requirements (ethical acceptance number: 201803674).

Data accessibility. Data have been uploaded as part of the electronic supplementary material.

Authors' contributions. B.L. designed the project, conducted experiments, analysed the data and wrote the manuscript; K.L. conducted experiments; Y.Q. designed the project and modified the manuscript. All authors had read and supported the submission of the manuscript.

Competing interests. We declare we have no competing interests.

Funding. The authors would like to appreciate National Natural Science Foundation of China (grant no. Project 21476265) and Graduate innovation project of Central South University (grant no. 1053320191828).

Acknowledgements. All authors are very grateful to all reviewers, and we gratefully acknowledge the help of Dr R. Fu of Xiangya Hospital, Central South University.

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
