## [Peer Review File · Royal Society Open Science]

Review History

RSOS-210462.R0 (Original submission)

Review form: Reviewer 1

Is the manuscript scientifically sound in its present form?

Yes

Are the interpretations and conclusions justified by the results?

Yes

Is the language acceptable?

Yes

Do you have any ethical concerns with this paper?

Yes

Have you any concerns about statistical analyses in this paper?

No

Recommendation?

Major revision is needed (please make suggestions in comments)

Comments to the Author(s)

General comments

This manuscript was reported that the development of polysulfone (PSf) membranes grafted with sulfonated citric chitosan (SCACS). This grafting may be more blood compatibility than conventional PSf membranes. There are many advantages to developing hemodialysis membranes with excellent blood compatibility. This paper will be helpful for physicians involved in hemodialysis.

However, these are some problems in this article. I suggest that the authors consider the following questions.

Major comments

1. Polyvinylpyrrolidone (PVP) already exists as a hydrophilizing agent with excellent blood compatibility. Many commercially available PSf membranes are hydrophilized by PVP. Therefore, the authors need to describe the reasons why SCACS is preferable to PVP in the manuscript.
2. Some type of coating on hemodialysis membranes may decrease solute clearance. Is the clearance of PSf membranes grafted with SCACS equivalent to that of ungrafted PSf membranes? If clearance has not been studied, this should be described in the limitation section.
3. It is described that the blood of a 25 year old female was used in "3.5 Hemocompatibility". However, it is assumed that bovine serum protein (BSA) was used for Protein adsorption (3.5.1). In "Platelet Adhesion and Transformation (3.5.2) - APTT, PT, TT (3.5.5)", was the blood of a 25 year old female used? It is difficult to understand, so please describe it clearly.
4. Please add statement on IRB approval for use with human blood (e.g. the approval number by the ethics committee).
5. Why is the sample size 3? How many samples were needed for this study? There is no sample size calculation in the methods section. The author should mention it in the methods section.

Minor comments

1. There are many spelling mistakes in the manuscript. e.g.: In line 28 page 2, "synthesised" is correctly "synthesized". In line 30 page 2, "achive" is correctly "achieve". In line 38 page 2, "droped" is correctly "dropped". There are many more spelling mistakes. Please correct them all.

Review form: Reviewer 2**Is the manuscript scientifically sound in its present form?**

No

Are the interpretations and conclusions justified by the results?

Yes

Is the language acceptable?

No

Do you have any ethical concerns with this paper?

No

Have you any concerns about statistical analyses in this paper?

No

Recommendation?

Major revision is needed (please make suggestions in comments)

Comments to the Author(s)

In this manuscript, sulfonated citric chitosan was grafted on PSF membrane to improve the hemocompatibility. Though the chemical structure and hemocompatibility of the modified membranes have been characterized, some comments are needed to be resolved. Thus, I think that a major revision is needed.

1. The language should be improved. For example, P5 Line 53 "AlCl₃"; P7 Line 35 "PSF-NH₂"; P6 Line 42 "Chloroacetylation"; P6 Line 43 "PSF1e"; P6 Line 37 "4 ml".
2. Can the PSF-CH₂Cl be fabricated into membranes without PSF? The group density is important to graft SCACS?
3. The result in Figure 9 shows similar platelet adhesion result for the modified and pristine membranes. In addition, the amounts of the platelets are needed to be counted.
4. Heparin-like polymers have been reported and used to modify membranes by reports such as *Macromol. Biosci.* 2020, 20, 2000153; *Biomacromolecules* 2016, 17, 4011-4020. Thus, the related references should be added and compared the results of blood compatibility.
5. To character the hemocompatibility, the result of SCACS is also important to analyze the function. What's more, statistical analysis for the data of APTT, TT and PT are needed.

Decision letter (RSOS-210462.R0)

Dear Dr Lin:

Title: Preparation of modified polysulfone material decorated by sulfonated citric chitosan for hemodialysis and its hemocompatibility
Manuscript ID: RSOS-210462

The editor assigned to your manuscript has now received comments from reviewers. We would like you to revise your paper in accordance with the referee and Subject Editor suggestions which

can be found below (not including confidential reports to the Editor). Please note this decision does not guarantee eventual acceptance.

Please submit your revised paper before 23-May-2021. Please note that the revision deadline will expire at 00.00am on this date. If we do not hear from you within this time then it will be assumed that the paper has been withdrawn. In exceptional circumstances, extensions may be possible if agreed with the Editorial Office in advance. We do not allow multiple rounds of revision so we urge you to make every effort to fully address all of the comments at this stage. If deemed necessary by the Editors, your manuscript will be sent back to one or more of the original reviewers for assessment. If the original reviewers are not available we may invite new reviewers.

RSC Associate Editor:
Comments to the Author:
(There are no comments.)

RSC Associate Editor:
Comments to the Author:
(There are no comments.)

Reviewers' Comments to Author:

Reviewer: 1

Comments to the Author(s)

General comments

This manuscript was reported that the development of polysulfone (PSf) membranes grafted with sulfonated citric chitosan (SCACS). This grafting may be more blood compatibility than conventional PSf membranes. There are many advantages to developing hemodialysis membranes with excellent blood compatibility. This paper will be helpful for physicians involved in hemodialysis.

However, these are some problems in this article. I suggest that the authors consider the following questions.

Major comments

1. Polyvinylpyrrolidone (PVP) already exists as a hydrophilizing agent with excellent blood compatibility. Many commercially available PSf membranes are hydrophilized by PVP. Therefore, the authors need to describe the reasons why SCACS is preferable to PVP in the manuscript.
2. Some type of coating on hemodialysis membranes may decrease solute clearance. Is the clearance of PSf membranes grafted with SCACS equivalent to that of ungrafted PSf membranes? If clearance has not been studied, this should be described in the limitation section.
3. It is described that the blood of a 25 year old female was used in "3.5 Hemocompatibility". However, it is assumed that bovine serum protein (BSA) was used for Protein adsorption (3.5.1). In "Platelet Adhesion and Transformation (3.5.2) - APTT, PT, TT (3.5.5)", was the blood of a 25 year old female used? It is difficult to understand, so please describe it clearly.
4. Please add statement on IRB approval for use with human blood (e.g. the approval number by the ethics committee).
5. Why is the sample size 3? How many samples were needed for this study? There is no sample size calculation in the methods section. The author should mention it in the methods section.

Minor comments

1. There are many spelling mistakes in the manuscript. e.g.: In line 28 page 2, "synthesised" is correctly "synthesized". In line 30 page 2, "achive" is correctly "achieve". In line 38 page 2, "droped" is correctly "dropped". There are many more spelling mistakes. Please correct them all.

Reviewer: 2

Comments to the Author(s)

In this manuscript, sulfonated citric chitosan was grafted on PSF membrane to improve the hemocompatibility. Though the chemical structure and hemocompatibility of the modified membranes have been characterized, some comments are needed to be resolved. Thus, I think that a major revision is needed.

1. The language should be improved. For example, P5 Line 53 "AlCl₃"; P7 Line 35 "PSF-NH₂"; P6 Line 42 "Chloroacetylation"; P6 Line 43 "PSF1e"; P6 Line 37 "4 ml".
2. Can the PSF-CH₂Cl be fabricated into membranes without PSF? The group density is important to graft SCACS?

3. The result in Figure 9 shows similar platelet adhesion result for the modified and pristine membranes. In addition, the amounts of the platelets are needed to be counted.
4. Heparin-like polymers have been reported and used to modify membranes by reports such as *Macromol. Biosci.* 2020, 20, 2000153; *Biomacromolecules* 2016, 17, 4011-4020. Thus, the related references should be added and compared the results of blood compatibility.
5. To character the hemocompatibility, the result of SCACS is also important to analyze the function. What's more, statistical analysis for the data of APTT, TT and PT are needed.

Author's Response to Decision Letter for (RSOS-210462.R0)

See Appendix A.

RSOS-210462.R1 (Revision)

Review form: Reviewer 1

Is the manuscript scientifically sound in its present form?

Yes

Are the interpretations and conclusions justified by the results?

Yes

Is the language acceptable?

Yes

Do you have any ethical concerns with this paper?

Yes

Have you any concerns about statistical analyses in this paper?

No

Recommendation?

Accept with minor revision (please list in comments)

Comments to the Author(s)

All the spelling mistakes that I previously suggested as minor comments have not been corrected. e.g.: In line 27 page 7/31, "achive" is correctly "achieve". In line 49 page 7/31, "treantment" is correctly "treatment". In line 27 page 8/31, "synthesised" is correctly "synthesized". There are many more spelling mistakes. Please correct them all.

Review form: Reviewer 2

Is the manuscript scientifically sound in its present form?

Yes

Are the interpretations and conclusions justified by the results?

Yes

Is the language acceptable?

Yes

Do you have any ethical concerns with this paper?

No

Have you any concerns about statistical analyses in this paper?

No

Recommendation?

Accept as is

Comments to the Author(s)

The manuscript now can accepted.

Decision letter (RSOS-210462.R1)

Dear Dr Lin:

Title: Preparation of modified polysulfone material decorated by sulfonated citric chitosan for hemodialysis and its hemocompatibility

Manuscript ID: RSOS-210462.R1

Thank you for submitting the above manuscript to Royal Society Open Science. On behalf of the Editors and the Royal Society of Chemistry, I am pleased to inform you that your manuscript will be accepted for publication in Royal Society Open Science subject to minor revision in accordance with the referee suggestions. Please find the reviewers' comments at the end of this email.

The reviewers and handling editors have recommended publication, but also suggest some minor revisions to your manuscript. Therefore, I invite you to respond to the comments and revise your manuscript.

Because the schedule for publication is very tight, it is a condition of publication that you submit the revised version of your manuscript before 30-Jun-2021. Please note that the revision deadline will expire at 00.00am on this date. If you do not think you will be able to meet this date please let me know immediately.

Kind regards,
Dr Laura Smith
Publishing Editor, Journals

RSC Associate Editor:
Comments to the Author:
(There are no comments.)

RSC Subject Editor:
Comments to the Author:
(There are no comments.)

Reviewer comments to Author:
Reviewer: 2
Comments to the Author(s)
The manuscript now can accepted.

Reviewer: 1
Comments to the Author(s)
All the spelling mistakes that I previously suggested as minor comments have not been corrected. e.g.: In line 27 page 7/31, "achive" is correctly "achieve". In line 49 page 7/31, "treantment" is correctly "treatment". In line 27 page 8/31, "synthesised" is correctly "synthesized". There are many more spelling mistakes. Please correct them all.

Author's Response to Decision Letter for (RSOS-210462.R1)

See Appendix B.

Decision letter (RSOS-210462.R2)

Dear Dr Lin:

Title: Preparation of modified polysulfone material decorated by sulfonated citric chitosan for hemodialysis and its hemocompatibility
Manuscript ID: RSOS-210462.R2

It is a pleasure to accept your manuscript in its current form for publication in Royal Society Open Science. The chemistry content of Royal Society Open Science is published in collaboration with the Royal Society of Chemistry.

RSC Associate Editor
Comments to the Author:
(There are no comments.)

Reviewer(s)' Comments to Author:

Appendix A

Response to the editor & reviewer's questions/remarks

Dear editor and reviewers,

Many thanks for your attention on our manuscript entitled “**Preparation of modified polysulfone material decorated by sulfonated citric chitosan for hemodialysis and its hemocompatibility**”(ID: **RSOS-210462**). We appreciate reviewers' useful comments and suggestions. The reviewers' comments are very helpful. We have carefully revised the manuscript according to the reviewers' comments, and marked with red color in manuscript. The following is a point-to-point response to the reviewers' comments. All the comments are marked with red color.

Reviewer#1:

This manuscript was reported that the development of polysulfone (PSf) membranes grafted with sulfonated citric chitosan (SCACS). This grafting may be more blood compatibility than conventional PSf membranes. There are many advantages to developing hemodialysis membranes with excellent blood compatibility. This paper will be helpful for physicians involved in hemodialysis. However, these are some problems in this article. I suggest that the authors consider the following questions.

Comment: 1) Polyvinylpyrrolidone (PVP) already exists as a hydrophilizing agent with excellent blood compatibility. Many commercially available PSf membranes are hydrophilized by PVP. Therefore, the authors need to describe the reasons why SCACS is preferable to PVP in the manuscript.

Response: Thank you for your kind reminder. PVP as a representative synthetic water-soluble polymer possesses excellent physiological inertia and biocompatibility, which has usually adopted as a hydrophilizing agent to enlarge the membrane pores of PSF for larger water flux in bleeding. However, PVP, as an hydrophilic material, is usually used for the modification by blending, and the PVP blend membranes can be uses for water treatment. However, for hemodialysis membrane, the stability is very important, and little PVP molecules may immerse in aqueous solution during ultrafiltration cause its solubility and weak affinity with PSF, which will bring potential risk for the nephrotic. In contrast, SCACS will not enter into blood during

hemodialysis because it is covalently grafted with PSF, furthermore, SCACS is an excellent modified anticoagulant with abundant hydrophilic groups as $-\text{SO}_3\text{H}$, $-\text{COOH}$ and $-\text{OH}$. Thus, SCACS was used to modify PSF.

Comment: 2) Some type of coating on hemodialysis membranes may decrease solute clearance. Is the clearance of PSf membranes grafted with SCACS equivalent to that of ungrafted PSf membranes? If clearance has not been studied, this should be described in the limitation section.

Response: The coating of the membrane surface is much thicker than the grafting modification of membrane surface, which is almost mono-molecule layer by chemical bond, and giving almost no change to the morphology of the membrane. But the modified PSF-SCACS membrane give much better biocompatibility than the original PSF membrane. That is to say, the rejection of some types of solutes is almost the same, but the adoption of protein and the thrombus is greatly decreased for PSF-SCACS membrane.

Comment: 3) It is described that the blood of a 25 year old female was used in “3.5 Hemocompatibility”. However, it is assumed that bovine serum protein (BSA) was used for Protein adsorption (3.5.1). In "Platelet Adhesion and Transformation (3.5.2) - APTT, PT, TT (3.5.5)”, was the blood of a 25 year old female used? It is difficult to understand, so please describe it clearly.

Response: Bovine serum protein (BSA) was usually used for Protein adsorption experiment, and it is also used in this work, and the blood compatibility tests, including platelet adhesion and transformation, APTT, PT, TT, etc, used real blood, and the blood of a 25 year old female was used. This is explained in the revised manuscript.

Comment: 4) Please add statement on IRB approval for use with human blood (e.g. the approval number by the ethics commmimttee).

Response: We have supplemented it in the part of Ethical Statement in the paper

according to the helpful suggestion.

Comment: 5) Why is the sample size 3? How many samples were needed for this study? There is no sample size calculation in the methods section. The author should mention it in the methods section.

Response: All the tested results were the average of three samples to eliminate the errors, and blood samples were 168 in this study. We have supplemented the above in the revised manuscript.

Reviewer #2

In this manuscript, sulfonated citric chitosan was grafted on PSF membrane to improve the hemocompatibility. Though the chemical structure and hemocompatibility of the modified membranes have been characterized, some comments are needed to be resolved. Thus, I think that a major revision is needed.

Comment: 1) The language should be improved. For example, P5 Line 53 “AlCl₃”; P7 Line 35 “PSF-NH₂”; P6 Line 42 “Chloroacetylation”; P6 Line 43” PSF1e”; P6 Line 37 ” 4 ml”.

Response: Thank you for your kind reminder, we have revised the whole manuscript carefully and tried to avoid any grammar or syntax error. In addition, we have asked several colleagues who are skillful in English to check the manuscript. These changes will not influence the content and framework of the paper. And here we did not list the changes but marked in red in the revised paper.

Comment: 2) Can the PSF-CH₂Cl be fabricated into membranes without PSF? The group density is important to graft SCACS?

Response: Thank you for your kind reminder. In our experiments, we found out that the PSF-COCH₂Cl with more active sites could still form a film by phase inversion after modification, but the resulting membrane has performed poorly in mechanical properties which prevents it far from the hemodialysis applications. Therefore, we have chosen a CAPSF/PSF blend membrane to ensure the mechanical performance and abundant active sites, and then we have found that the mechanical properties of

the membranes were basically the same when PSF-COCH₂Cl (CAPSF) was 8wt% and PSF was 10wt% in bleeding membranes.

For further modification, the CAPSF membrane is the bridge to connect the inert skeleton of PSF and anticoagulant factor SCACS, which can effect the final grafting rate of SCACS for further hydrophilicity and anticoagulant. Thus, we have worked variable control of reaction temperature and time to find the optimal reaction conditions of CAPSF for abundant active sites and higher grafting rate of SCACS.

Comment: 3) The result in Figure 9 shows similar platelet adhesion result for the modified and pristine membranes. In addition, the amounts of the platelets are needed to be counted.

Response: Thank you for your kind reminder, we are really sorry that the SEM images of platelets adsorbing onto pristine and modified PSF membranes were selected incorrectly in Figure 9, but the SEM images of platelets adsorbed on pristine and modified PSF membranes have been corrected, as shown in Figure 9 (a).

The platelet adhesion as a biological phenomenon of contacting behaviors between the blood and materials is carried out in vitro simulated blood environment, which could partly indicate the change of blood compatibility between the original and modified membranes. The SEM images of platelets could show the number and shape of platelets onto the membranes for intuitive changes of blood compatibility onto the pristine and modified membranes. Among the images, the amounts of platelets are unable to be calculated clearly, but the aggregation degree could clearly point at the adhesion degree of platelets on the membrane. Simultaneously, there are so few reports on the calculation of the amounts of platelets that it is difficult to achieve this work by a precise method. We are sorry that we may not be able to complete the quantity measurement of platelets in a short time. We all appreciated this helpful suggestion, and we will continue to try to find an accurate method to quantitatively characterize platelet adsorption in future research work.

Comment: 4) Heparin-like polymers have been reported and used to modify

membranes by reports such as *Macromol. Biosci.* 2020, 20, 2000153; *Biomacromolecules* 2016, 17, 4011-4020. Thus, the related references should be added and compared the results of blood compatibility.

Response: We have cited the above literature and briefly compared the results of blood compatibility in introduction section.

Comment: 5) To character the hemocompatibility, the result of SCACS is also important to analyze the function. What's more, statistical analysis for the data of APTT, TT and PT are needed.

Response: Thank you for your kind suggestion, we have done briefly the statistical analysis for the data of APTT, TT and PT according to the reviewer's comment, and the results have been shown in Figure 10.

Appendix B

Response to the editor & reviewer's questions/remarks

Dear Dr Laura Smith and reviewers,

Many thanks for your attention on our manuscript entitled “**Preparation of modified polysulfone material decorated by sulfonated citric chitosan for hemodialysis and its hemocompatibility**”(ID: **RSOS-210462**). We appreciate reviewers' useful comments and suggestions. The reviewers' comments are very helpful. We have carefully revised the manuscript according to the reviewers' comments, and marked with red color in manuscript. The following is a point-to-point response to the reviewers' comments. All the comments are marked with red color.

Reviewer#1:

Comment: 1) All the spelling mistakes that I previously suggested as minor comments have not been corrected. e.g.: In line 27 page 7/31, “achive” is correctly “achieve”. In line 49 page 7/31, “treantment” is correctly “treatment”. In line 27 page 8/31, “synthesised” is correctly “synthesized”. There are many more spelling mistakes. Please correct them all.

Response: Thank you for your kind reminder, we have revised the whole manuscript carefully and tried to avoid any grammar or syntax error. In addition, we have asked several colleagues who are proficient in English to check the manuscript. These changes will not influence the content and framework of the paper. And here we did not list the changes but marked in red in the revised paper.